# Working life sequences over the life course among 9269 women and men in Sweden; a prospective cohort study

**Katalin Gémes**[1]*, **Katriina Heikkilä**[1,2], **Kristina Alexanderson**[1], **Kristin Farrants**[1], **Ellenor Mittendorfer-Rutz**[1], **Marianna Virtanen**[1,3]

**1** Division of Insurance Medicine, Department of Clinical Neuroscience, Karolinska Institutet, Stockholm, Sweden, **2** Population Health Unit, Finnish Institute of Health and Welfare, Helsinki, Finland, **3** School of Educational Sciences and Psychology, University of Eastern Finland, Joensuu, Finland

* katalin.gemes@ki.se

## Abstract

**Data Availability Statement:** These data cannot be made publicly available due to privacy regulations. According to the General Data Protection Regulation, the Swedish law SFS 2018:218, the

### Objectives

To investigate working life courses in women and men and possible associations with socio-economic, health-, and work-related factors.

### Methods

A 15-year prospective cohort study of individuals aged 18–50 in paid work at baseline and answering the Swedish Living Conditions Surveys (2000–2003, N = 9269) and their annual economic activity, using nationwide registers. We used sequence and cluster analyses to identify and group similar working life sequences. Multinomial logistic regression was used to examine associations of sex, socioeconomic, health-, and work-related factors with sequence cluster memberships.

### Results

We identified 1284 working life sequences, of which 65% represented continuous active (in paid work/studying) states. We then identified five sequence clusters, the largest one with individuals who were continuously active (n = 6034, 65% of the participants; 54% of women and 76% of men) and smaller ones with interruptions of the active state by long-term paren-tal-leave, unemployment, and/or sickness absence/disability pension (SA/DP), or retire-ment. Women were more likely than men to belong to the "Parental-leave periods" (odds ratio [OR]: 33.2; 95% confidence intervals [CI]: 25.6, 43.1) and the "SA/DP periods" sequence clusters (OR: 1.8; 95% CI: 1.4, 2.1), also after adjustment for covariates. In both sexes, low education and poor health were the strongest predictors of belonging to the sequence cluster "Unemployment & SA/DP periods". Predictors of the "Parental-leave peri-ods" sequence cluster differed between women and men.

Swedish Data Protection Act, the Swedish Ethical Review Act, and the Public Access to Information and Secrecy Act, these types of sensitive data can only be made available for specific purposes, including research, that meets the criteria for access to this type of sensitive and confidential data as determined by a legal review. Readers may contact Professor Kristina Alexanderson (kristina. alexanderson@ki.se) and Mira Muller (mira. muller@ki.se) regarding the data.

**Funding:** This work was financially supported by a research grant from the Swedish Research Council for Health, Working Life and Welfare, FORTE [grant number 2018-00547, receiver: MV]. We utilised data from the REWHARD consortium supported by the Swedish Research Council (grant number 2017-00624). There was no additional external funding received for this study. The funders had no role in study design, data collection and analysis, decision to publish, or preparation of the manuscript.

**Competing interests:** The authors have declared that no competing interests exist. Heikkilä Katriina and Ellenor Mittendorfer-Rutz are Academic Editor at the Journal

## Conclusions

In a cohort of individuals in paid work at baseline, the majority of women and men worked most of each year although women were more likely to have some interruptions characterized by long-term parental-leave or SA/DP periods than men, independently of socioeconomic, health-, and work-related factors.

## Introduction

In high-income countries, increased life expectancy and low birth rates have led to an increase in the old-age dependency ratio, i.e., there are fewer workers in the labour market to cover the increasing healthcare and social security costs [1]. Many governments have started to reform public pension systems in order to extend working lives and avoid 'the pensions crisis', e.g., by raising the retirement age. Extending working life, and ensuring good health and active participation in the labour market for all are crucial to succeeding with such policy changes [2]. However, there is a gap between women and men in the duration of working life; men seem to work more years than women and women seem to have lower working life participation rates throughout the life course [2, 3] This can be related to various reasons, including that women often spend longer time with unpaid work, such as caring for children or other family members, have higher morbidity during working age, or take old-age pension earlier than men [2, 4]. Furthermore, more women work in temporary or part-time jobs, often jobs characterised by unstable and insecure labour market positions [2, 5, 6] compared to men. Furthermore, women's jobs are more likely to be characterised by poorer career prospects, higher psychosocial workload [7, 8], poorer control over working hours [9], and higher rate of workplace bullying and harassment [10, 11]. On the other hand, men more often work longer hours, report higher effort-reward imbalance, lower social support, and higher physical demands at work [7]. Many of these work characteristics have been shown to be associated with interruptions in working-life participation or early exit from the labour market by means of long-term sickness absence (SA) and disability pension (DP) [10–15]. However, studies mapping the relationship between socioeconomic, health-, work- related factors and diverse working life courses, for example, those including longer health-, work-, and family-related career interruptions, are scarce. A major reason for this is the limited availability of good quality data which can combine information from representative surveys and administrative registers. The assessment of perceived physical and psychosocial work-characteristics, self-rated health, and lifestyle requires survey-based methods while administrative registers are important sources of information on annual states regarding employment-, family and health related benefits, without the problem of drop out during the follow-up.

Some of the previous research focused on the examination of the inter- and intra-individual diversity of work-family life courses [3, 16–20]. McMunn at al., [16] examined the changes in work-family life courses over time and showed that women's work-family life courses converged to men's while the diversity of the individual work-family life courses increased. They also described that those work-family life courses which indicated a weaker link to paid work, e.g., having early childbirth without employment, which was more common in women than men, was found to be a predictor of poor mental health [17], lower subjective well-being [18], lower quality of life [19], worse body mass index trajectories [20], and also a lower probability of later employment [3]. However, as the focus was on the consequences of work-family life courses, these studies did not include health-related breaks in their assessment, such as long-term SA and DP. According to the integrated life-course perspective proposed by Amick et al.

[21], both labour market and health transitions create and shape an individual's working life course. Health-, work-, and family-related states and transitions are interrelated and the individual's working life course takes place within the labour market and social contexts.

Sequence analysis is an advanced analytical tool to capture the complexity of working life courses [22, 23] and it has been used in previous research in relation to work-family life courses [3, 17–20], Therefore, we will use this analytical approach in our study to examine sequences of family-, work-, and health-related life courses simultaneously. In line with the integrated life course perspective, we aimed to map 15 years of working life courses among individuals aged 18 to 50 and in paid work at inclusion, and among them, to identify groups with similar working life sequences, and to examine whether sex was associated with working life sequences, independently of socioeconomic, health-, and work-related conditions. Furthermore, we aimed to examine which of these factors were associated with the membership of specific working-life sequence groups separately among women and men. The specific research questions were to explore: 1) which types of overarching future working life sequence groups can be identified during a 15-year period in a cohort of individuals in paid work at baseline? 2) whether women have more often than men working life courses characterized by several interruptions, such as parental leave, SA/DP, or unemployment? 3) whether adjusting for baseline sociodemographic, health- and work-related factors weaken the possible associations between sex and the specific sequence cluster membership? 4) whether these factors are independently associated with the specific sequence cluster membership in a separate analysis among women and men?

## Materials and methods

The research was approved by the Regional Ethical Review Board, Stockholm, Sweden, who also waived the need for informed consent [24, 25].

A prospective cohort study was conducted using microdata from several sources. First, the Living Conditions Surveys (ULF/later SILC: *Undersökningarna av levnadsförhållanden/EU Statistics on Income and Living Conditions*) [26] was used. This is an annual, mainly phone-based survey by Statistics Sweden on a random sample of individuals aged 16 to 84 and living in Sweden. We used data from the surveys of 2000, 2001, 2002, and 2003. Overall, 5678 responded in 2000 (76% of invited), 5805 in 2001 (78% of invited), 6322 in 2002 (75% of invited), and 6363 in 2003 (75% of invited). In our study, we included respondents who were aged 18–50 years at survey date, who were in paid work (i.e., full- or part-time employed or self-employed), and had been living in Sweden for at least two years, i.e., the survey year and the preceding year, and had no missing information on variables we were interested in (S1 Fig).

Statistics Sweden linked the survey data to microdata from nationwide registers, using the unique identification numbers assigned to all registered residents in Sweden [27], and anonymized the dataset. Register data were obtained for the year before the survey and annually for up to 15 years after. Follow-up began at the beginning of the year following the survey year. Individuals who emigrated from Sweden or died during the follow-up period (n = 252) were excluded (S1 Fig).

### Survey data

Economic hardship (having difficulties in making ends meet during the last year: yes; no), working conditions such as number of weekly working hours (≤35; >35 and ≤45; >45 hours per week), exposure to noise at work (yes, to medium-strong noise for substantial amount of time; no or only light noise), hectic work schedule (yes; no), mentally strenuous job (yes; no), physically strenuous job (yes; no), possibility to learn new things at work (yes; no), and having

had a work accident in the past 12 months (yes; no). Health and health-related behaviors included self-rated health (good/average; poor), having long-term illness or long-term health problem (yes; no), smoking (daily smoker; not a daily smoker), overweight/obesity (body mass index, BMI $\leq 25$ kg/m$^2$; $>25$ kg/m$^2$).

## Register data

Baseline sociodemographic variables, annual labor market status, and year of emigration were obtained from the Longitudinal Integrated Database for Health Insurance and Labor Market Studies (LISA) held by Statistic Sweden [28]. Information on sickness absence/disability pension (SA/DP) spells (start and end dates and yearly net days) were obtained from the Micro-Data for Analysis of the Social Insurance System (MiDAS) held by the Social Insurance Agency [29]. Year of death from the Cause of Death Register held by the National Board of Health and Welfare.

We considered the following register-based sociodemographic factors at baseline: sex (woman; man), age ($<31$; $31$–$40$; $>40$ years), nationality (Swedish; not Swedish), both parents/participants themselves being born outside Sweden (yes/no); educational level (compulsory; secondary; university/college), type of living area (Stockholm; Gothenburg/Malmö; other large cities; middle size cities, small towns and rural areas), family situation (single without children aged $<18$ years; married or cohabitating without children aged $<18$ years; single with children aged $<18$ years; married or cohabiting and living with children aged $<18$ years), and a history of long-term SA/DP ($>183$ net days of SA and/or DP during the survey year or the preceding year: yes/no).

The following mutually exclusive annual labor market states were defined: *"active"* if the participant received income from work or student benefit and was not categorized to the other states;*"unemployed"* if the participant had unemployment benefit $>183$ days/year; *"on parental leave"* if the participant had parental-leave benefits $>183$ days/year; *"on SA/DP"* if the participant had SA and/or DP benefits $>183$ net days/year; and *"retired"* if $>50\%$ of the annual income was from old-age pension.

## Social security in Sweden

In Sweden, people aged at least 16 years and having income from work, unemployment benefits, or parental-leave benefits are eligible for SA benefit if morbidity leads to work incapacity. The first SA day is a waiting day without compensation; from the eighth day of a SA spell, a medical certificate from the treating physician is required [30]. For employed people, the employer pays the SA benefit for days 2–14, after that, the Swedish Social Insurance Agency pays the benefit. For the unemployed, this happens from the second day of the SA spell. In order to not to introduce bias, we used information on SA spells $>14$ days. All people in Sweden aged 19–64 can be granted DP if morbidity has led to long-term or permanent reduction of work capacity. Both SA and DP can be granted for full- or part-time of ordinary work hours. Everyone over the age of 16, capable of working, and registered as a jobseeker is eligible for an unemployment benefit [31]. Sweden, similar to other Nordic countries, follows the dual-earner family policy model where both parents are supposed to be involved in both labour market activity and domestic work, and both parents are entitled to extensive number of days with parental-leave benefits [32]. In the years studied, parental-leave benefits could be obtained for 480 days per child until the child was ten years. Some of these days were earmarked for each respective parent, for the rest of the days, the parents could decide how to share [32]. Tax-financed child care is available for children aged 1 to 10 years for a small, income-based fee. The median age at the birth of the first child in Sweden was 28.2 for women

and 30.7 for men in 2000, during the first survey year and it was 29.2 and 32.5, respectively, in 2016, the last follow-up year. In 2000, the fertility rate in Sweden was 1.5 children and increased to 1.8 by 2016 [33]. There is no fixed retirement age in Sweden although in the years studied, one could apply for old-age pension from the age of 61; the usual retirement age was 65 years.

## Statistical analysis

Differences between women and men regarding socioeconomic, work, and health-related factors were described by calculating proportions and chi$^2$ statistics. We then conducted sequence analyses to identify temporal sequences of annual labour market activity states and to group them into empirically distinct sequence clusters. A sequence was defined as an ordered list of elements (here, several possible labour market activity states on a yearly basis) [22] over a period of 15 years. The relative proportion of each of the five states for each year was displayed in state proportion plots. The mean duration of years within a given labour market activity state and the mean number of episodes of different states were also calculated. To group similar sequences, we conducted weighted cluster analyses. First, we calculated dissimilarity measures between the individual sequences by using optimal matching algorithms defined by the cost of transforming one sequence into another [21, 22, 34]. For the dissimilarity measures, a cost of one was assigned to insertion and deletion of states, and for substitution, the cost of transition was based on values of the calculated transition matrix. Rare transitions contributed to a higher cost, and frequent transitions to a lower cost. Second, we used hierarchical agglomerative cluster algorithm with Ward's linkage to grouped similar sequences to clusters [21, 22, 34]. Third, we evaluated the resulting 2–10 cluster solutions by using several partition quality measures listed in S1 Table and by looking at the hierarchical cluster tree diagrams of the resulting numbers and membership sizes of the clusters and the suggested sequence patterns. We also tested the constant substitution cost and partition around medoids (PAM) clustering algorithms, to see whether they resulted in similar cluster solutions.

After deciding the optimal sequence cluster solution, assigning all individuals to one of the clusters, and creating sequence index plots by clusters, we used multinomial logistic regression analysis to examine the associations between sex and the sequence cluster membership. To examine whether the associations between sex and cluster memberships attenuated, several sets of covariates were gradually included in the models. Model 1 was adjusted for age only, model 2 additionally included all socioeconomics; model 3, included model 2 plus the health-related covariates, and model 4 included model 3 plus the work-related covariates. We calculated odds ratios (OR) with 95% confidence intervals (CI) to compare sequence cluster memberships, using men and the cluster with the largest number of individuals as the reference categories. Thereafter, we investigated the associations between covariates and sequence cluster memberships separately among women and men. In sensitivity analyses we showed other possible cluster solutions a solution for one cluster less and one cluster more compared to the chosen one. To explore whether these cluster solutions were similar among women and men, we presented the results from cluster analysis stratified by sex. Furthermore, we ran the multivariable adjusted analysis between sex and cluster memberships with these alternative cluster solutions. We also reran the main analysis by adjusting the models for those variables at baseline that can be assumed to be stable over time for the majority of individuals, such as education, baseline age, birth country, nationality, living area and SA/DP history.

Data management was performed with SAS (version: 9.4) and statistical analysis with R (version 4.0.5) using the TraMineR [35] and WeightedCluster [34] packages.

## Results

Descriptive statistics of the study variables are provided in Table 1. Compared with men, women, on average, had higher education, were more often single with children, worked shorter hours, were less likely to have physically strenuous work and more likely to have mentally strenuous work.

We identified 1284 unique future labour market sequences among the participants (Figs 1, 2). The sequence with the largest number of participants included *"active"* states during the whole follow-up period (65.1%). Other common sequences were *"active"* states alternating with *"on parental leave"*, *"unemployed"*, and *"on SA/DP"* states (Figs 1, 2 and S2, S3 Figs). With regard to transitions to the *"active"* state, the most likely transitions were staying *"active"* (*"active"*-*"active"*; 97%) and returning to an *"active"* state after *"on parental leave"* state (*"on parental leave"*-*"active"*; 83%) (S2 Table). The lowest probability of transition was returning to the *"active"* state from the *"retired"* state (0.1%) (S2 Table).

The cluster partition quality measures showed the best fits for a five- and a seven-cluster solution (lowest point biseral correlation for five clusters and highest Hubert's Gamma, Average Silhouette Width, and Calinszki-Habarasz indexes for the seven-clusters solution) but only with a minimal difference in indexes between the five- and seven-cluster solutions (S1 Table, S4 and S5 Figs). To consider the feasibility and readability of the multinomial regression analyses, we chose the five-sequence cluster solutions for further analysis (Fig 3). In the five-cluster solution, the mean time spent in each state is presented in S6 Fig. Four-, six-, and seven-cluster solutions are presented in S7–S9 Figs. Most participants (n = 6034; 65.1%; 53.9% of women, 75.8% of men) belonged to the "Active" cluster, which comprised exclusively *"active"* state sequences during the entire follow-up. Sequences which combined mostly *"active"* states with some *"on parental leave"* states were also identified as a cluster (n = 1179; 12.7%; 24.4% of women and 1.6% of men), and this was named as "Parental-leave periods". The third largest cluster included a mix of sequences, combining mostly *"active"* states with *"unemployed"*, and *"on SA/DP"* states and was named "Unemployment & SA/DP periods" cluster (n = 1018; 11.0%; 10.0% of women and 12.5% of men). Finally, we identified two relatively small clusters, one with sequences including *"on SA/DP"* states besides *"active"* states (n = 710; 7.7%; 9.1% of women and 6.3% of men), therefore, named "SA/DP periods" cluster and one with sequences including *"active"* and *"retired"* sequence states (n = 328; 3.5%; 3.2% of women and 3.8% of men), and named as "Retirement". Using other types of distance measures and cluster algorithms resulted in similar cluster distributions as was found with a five-cluster solution that showed a good cluster partition quality and identified similar clusters.

When the cluster analysis was stratified by sex, we found a four-cluster solution to have the best fit for women and a four- and five-clusters solutions to have the best fit for men (S3 Table, S10 Fig). We present the four clusters solutions for both women and men with mean time spent in each state (S11–S14 Figs).

The results regarding the association between sex and cluster membership are presented in Table 2. The "Active" cluster was used as the reference category and the ORs and 95% CIs for the membership of another cluster were calculated for women, with men as the reference category. Adjustment for different sets of variables did not change the estimates considerably. After adjusting for all covariates, women were still 33.2 times more likely to belong to the "Parental-leave periods" cluster (95% CI: 25.6 to 43.1) and 1.8 times more likely to belong to the "SA/DP periods" cluster than men (95% CI: 1.4 to 2.1). There was no sex difference regarding the memberships of the "Unemployment & SA/DP periods" (adjusted OR: 1.0, 95% CI: 0.9 to 1.2) and the "Retirement" cluster (adjusted OR: 0.9, 95% CI: 0.7 to 1.1). When the models were adjusted for covariates that were time-fixed, we got similar estimates (adjusted OR for "Parental

**Table 1. Baseline characteristic of the study participants.**

| | All (n = 9269) | Women(n = 4516) | Men(n = 4753) | p* |
|---|---|---|---|---|
| **Sociodemographic, socioeconomic (n, (%))** | | | | |
| Age | | | | |
| <31 years | 2760 (29.8) | 1293 (28.6) | 1467 (30.9) | 0.01 |
| 31–40 years | 3232 (34.9) | 1562 (34.6) | 1670 (35.1) | |
| >40 years | 3277 (35.4) | 1661 (36.8) | 1616 (34.0) | |
| Educational level | | | | |
| Compulsory | 1058 (11.4) | 409 (9.1) | 649 (13.7) | <0.01 |
| Secondary | 4997 (53.9) | 2347 (52.0) | 2650 (55.8) | |
| University, college | 3214 (34.7) | 1760 (39.0) | 1454 (30.6) | |
| Type of living area | | | | |
| Stockholm | 1900 (20.5) | 946 (20.9) | 954 (20.1) | 0.08 |
| Gothenburg/Malmö | 1468 (15.8) | 729 (16.1) | 739 (15.5) | |
| Other larger cities | 3335 (36.0) | 1647 (36.5) | 1688 (35.5) | |
| Small, middle-sized cities, rural | 2566 (27.7) | 1194 (26.4) | 1372 (28.9) | |
| Family situation | | | | |
| Living alone without children | 1865 (20.1) | 664 (14.7) | 1201 (25.3) | <0.01 |
| Married/cohabiting with children <18 years | 4503 (48.6) | 2275 (50.4) | 2228 (46.9) | |
| Married/cohabiting without children | 2314 (25.0) | 1151 (25.5) | 1163 (24.5) | |
| Living alone with children <18 years | 587 (6.3) | 426 (9.4) | 161 (3.4) | |
| Not Swedish citizen | 387 (4.2) | 199 (4.4) | 188 (4.0) | 0.30 |
| Both parents/participants themselves being born outside Sweden | 1201 (13.0) | 608 (13.5) | 593 (12.5) | 0.17 |
| Experiencing economic hardship | 1484 (16.0) | 757 (16.8) | 727 (15.3) | 0.06 |
| **Health-related factors (n (%))** | | | | |
| Poor self-rated health | 232 (2.5) | 122 (2.7) | 110 (2.3) | 0.26 |
| Previous SA/DP >183 net days | 191 (2.1) | 118 (2.6) | 73 (1.5) | <0.01 |
| Long-term illness or health-problem | 3145 (33.9) | 1598 (35.4) | 1547 (32.5) | <0.01 |
| Daily smoker | 1609 (17.4) | 913 (20.2) | 696 (14.6) | <0.01 |
| Overweight/obese (BMI >25 kg/m$^2$) | 3608 (38.9) | 1284 (28.4) | 2324 (48.9) | <0.01 |
| **Working conditions (n (%))** | | | | |
| Weekly working hours | | | | |
| 36–44 | 6130 (66.1) | 2643 (58.5) | 3487 (73.4) | <0.01 |
| ≤35 | 2067 (22.3) | 1634 (36.2) | 433 (9.1) | |
| ≥45 | 1072 (11.6) | 239 (5.3) | 833 (17.5) | |
| Physically strenuous work | 2512 (27.1) | 1001 (22.2) | 1511 (31.8) | <0.01 |
| Exposed to noise | 2614 (28.2) | 870 (19.3) | 1744 (36.7) | <0.01 |
| Mentally strenuous job | 4268 (46.0) | 2310 (51.2) | 1958 (41.2) | <0.01 |
| Monotonous job | 3852 (41.6) | 1940 (43.0) | 1912 (40.2) | 0.01 |
| Hectic work | 6667 (71.9) | 3317 (73.4) | 3350 (70.5) | <0.01 |
| Little/no opportunity to learn new things at work | 6650 (71.7) | 3154 (69.8) | 3496 (73.6) | <0.01 |
| Work-related accident | 776 (8.4) | 298 (6.6) | 478 (10.1) | <0.01 |

*P-value for difference in $\chi^2$-test for categorical variables and two-tailed Student's t-test for age.SA: Sickness absence; DP: disability pension

leave cluster": 32.1, 95% CI:25.0 to 41.2; adjusted OR for "SA/DP periods": 1.8, 95% CI: 1.5 to 2.1; adjusted OR for "Unemployment & SA/DP periods": 1.1, 95% CI: 1.0 to 1.3; and adjusted OR for "Retirement" was 1.0, 95% CI: 0.8 to 1.2). The results of the baseline and multivariable-adjusted sensitivity analyses for the four and six clusters solutions are presented in S4 Table.

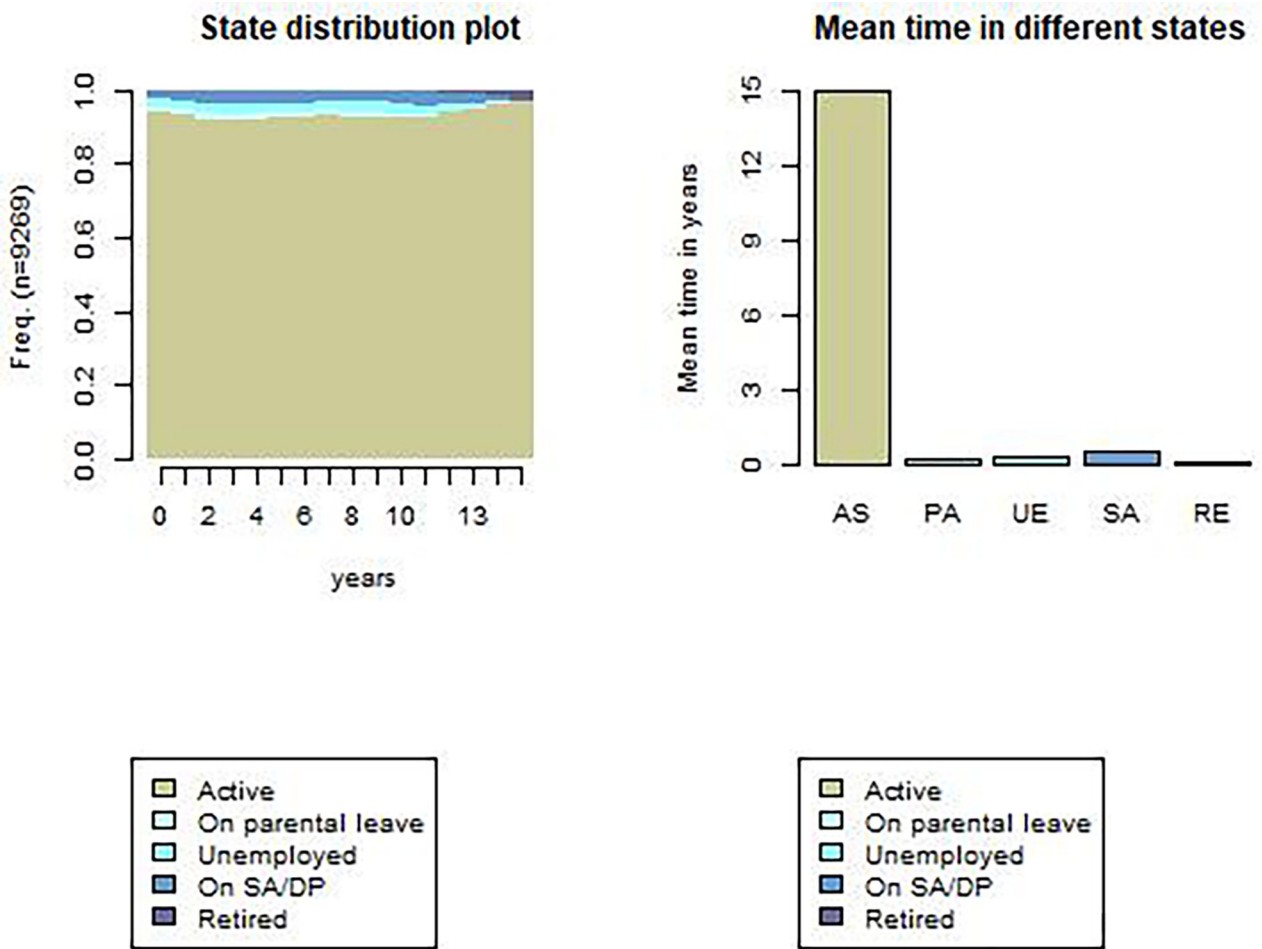

**Fig 1. State distribution plots and mean time in different states of future yearly working life sequences over a 15-year period.**

We then stratified our analysis by sex and investigated which variables were associated with different cluster memberships among women and men separately, again using the "Active" sequence cluster as the reference category. Lower education, living outside of the Stockholm area, living alone without children, economic hardship, both parents/participants themselves being born outside of Sweden, and variables indicating poor health and health-behaviours were associated with the "Unemployment & SA/DP periods" clusters membership in both sexes (S5 and S6 Tables). Among women, younger age (S5 Table) and among men (S6 Table), older age, working <35 hours per week (compared with ≥35 and <45 hours) and some work-related factors were associated with the "Unemployment & SA/DP periods" cluster membership. Younger age was in both sexes associated with membership of the "Parental-leave periods" cluster. Higher education, economic hardship, some health-related factors, working <35 hours per week (compared with ≥35 and <45 hours), and other work-related factors were associated with membership of the "Parental-leave periods" cluster among women, while among men, living in Stockholm, cohabiting with a partner and children, and being on SA in the previous year were associated with the "Parental-leave periods" cluster membership. Belonging to the "SA/DP periods" cluster was strongly associated with a history of SA/DP among both sexes as well as with older age, lower education, poor health and health related factors. Living outside the Stockholm area, living alone with

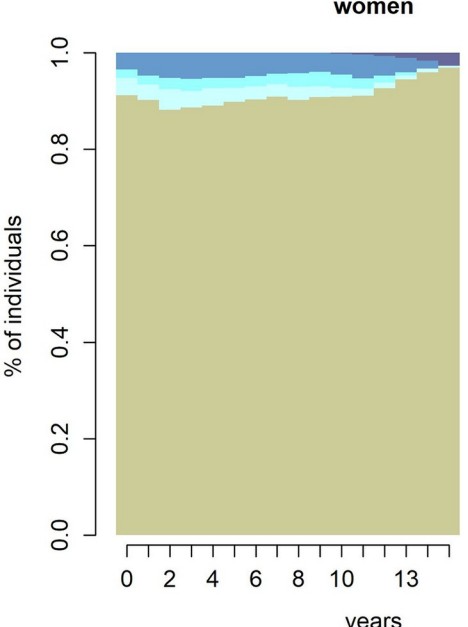
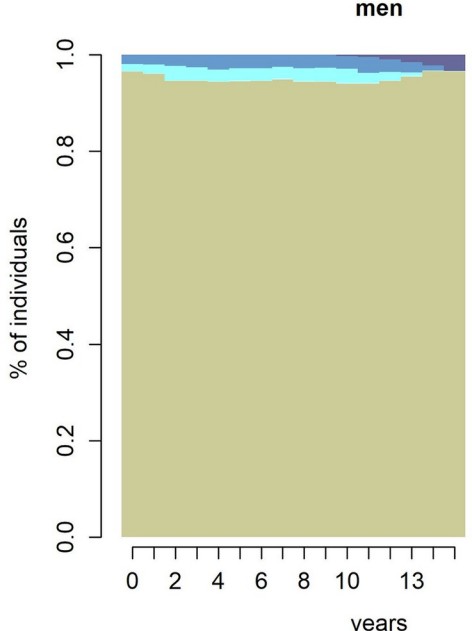

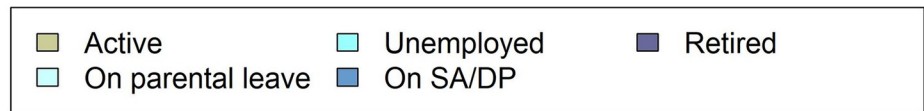

**Fig 2. State distribution plots of future yearly working life sequences over a 15-year period, among women and men, respectively.**

children, either participants themselves or their parents being born outside of Sweden, and some work-related factors were associated with the "SA/DP periods" cluster membership among women, and either participants themselves or their parents being born outside of Sweden and living alone among men. Older age and history of SA/DP were associated with belonging to the "Retirement" cluster among women while older age, being married/cohabiting without children, and poor health were associated with the "Retirement" cluster membership among men.

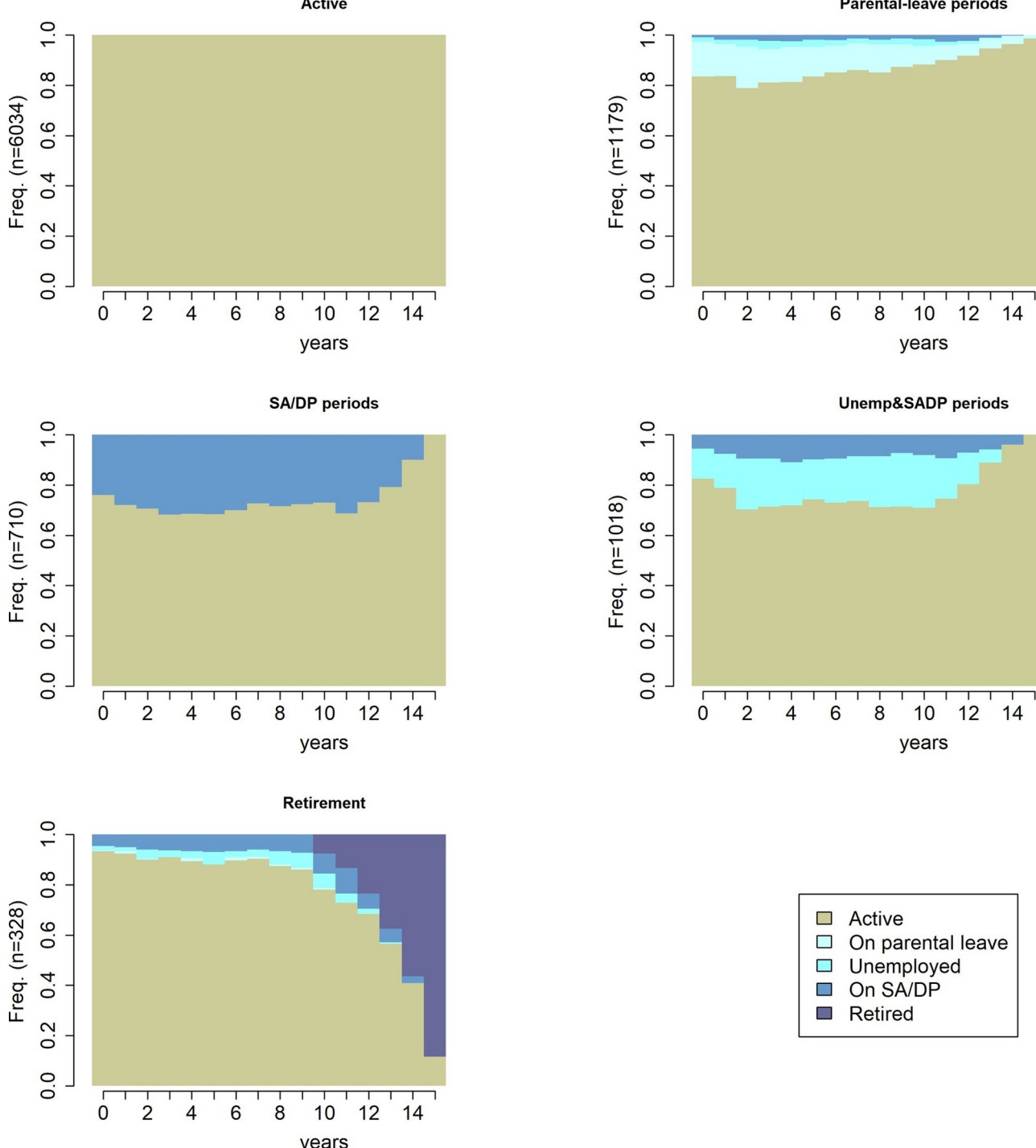

**Fig 3. Sequence index plots of the five clusters of working life sequences.**

## Discussion

In this prospective cohort study of a random sample of individuals initially in paid work in Sweden, we identified a total of five clusters of working life sequences over the 15-year follow-

**Table 2. Multinomial regression analysis for the associations between sex and different working life sequence clusters.**

| Cluster name | Number of individuals | Model 1 | Model 2 | Model 3 | Model 4 |
|---|---|---|---|---|---|
| | | OR (95% CI) for women compared with men | OR (95%CI) for women compared with men | OR (95%CI) for women compared with men | OR (95%CI) for women compared with men |
| Active | 6034 | Ref. | Ref. | Ref. | Ref. |
| Unemployment & SA/DP periods | 1018 | 1.1 (0.9, 1.3) | 1.2 (1.0, 1.4) | 1.2 (1.0, 1.3) | 1.0 (0.9, 1.2) |
| Parental leave periods | 1179 | 33.3 (26.0, 42.7) | 30.7 (23.9, 39.4) | 32.1 (24.9, 41.4) | 33.2 (25.6, 43.1) |
| SA/DP periods | 710 | 1.8 (1.5, 2.1) | 1.9 (1.6, 2.2) | 1.9 (1.6, 2.2) | 1.8 (1.4, 2.1) |
| Retirement | 328 | 0.9 (0.8, 1.2) | 1.0 (0.8, 1.2) | 1.0 (0.8, 1.2) | 0.9 (0.7, 1.1) |

SA/DP: sickness absence, disability pension; OR: odds ratio; CI: confidence interval

Model 1: adjusted for age.

Model 2: model 1 + education, both parents/participants themselves was born outside Sweden, nationality, family situation, living area, economic hardship.

Model 3: model 2 + previous SA/DP, long-term illness or health-problem, daily smoking and overweight/obesity.

Model 4: model 3 + working hours, physically strenuous work, monotonicity, noise exposure, mentally strenuous job, hectic work schedule, opportunity to learn new things and job accidents.

up. The vast majority (65%; 54% of women and 76% of men) belonged to the "Active" sequence cluster in which employment or studying was the exclusive labour market activity state. Other clusters combined mostly *"active"* states with *"on parental leave"*, *"unemployed"*, *"on SA/DP"*, and *"retired"* states. Women were 33.2 times more likely than men to belong to the "Parental-leave periods" and 1.8 times more likely to the "SA/DP periods" clusters than men, and these associations were only to a small degree affected by adjustments for sociodemographic factors, living conditions, health, and work-related factors.

Our finding that women were more likely than men to have working life patterns characterised by long parental-leave periods is in line with the studies from the U.S. and Finland [36–38]. Finland has a similar welfare state model as Sweden [39], encouraging equal contribution to childcare activities among the parents, providing good quality, public, low-fee childcare, and promoting equal participation in the labour market. However, similar sex differences as in our study were found in Finland in the early career trajectories even among those with a stable labour market attachment [37]. A U.S. study from the National Longitudinal Survey of Youth showed that women were more likely to have unstable employment trajectories and lower level of employment over their life course than men [36]. In Sweden, long parental leave among women has been associated with a lower likelihood of upward occupational mobility [40].

In this cohort of people in paid work, women were not more likely than men to end up in working life sequences characterized by long-term unemployment or retirement. Instead, we found that women were slightly more likely to belong to a sequence with longer SA/DP periods. While several work- and non-work-related factors at baseline were adjusted for, it is still possible that some of these factors changed over time, which we could not consider, e.g., stress from work and domestic activities, which might lead to higher SA/DP rates among women. Sex differences in disease susceptibility and incidence for diseases such as common mental disorders with a higher incidence among women than men, may also have contributed to the observed differences. A Finnish register-based study with a 20-year follow-up found no sex difference in the long-term risk of DP following the first SA spell [41]. However, the outcome of interest was DP, while we measured both SA and DP with different lengths and sequences over the working life course, and therefore, we were also able to capture individuals with longer, but still temporary SA, with the possibility of later returning to the labour market.

When we analysed the association between baseline characteristics and cluster membership among women and men separately, we found that overall, the same characteristics predicted membership in the "Unemployment & SA/DP periods" sequence clusters in both sexes. Lower education, either participants themselves or their parents born outside of Sweden, economic hardship, living outside of Stockholm, and poor health were associated with this cluster membership. This is in line with the findings of a recent review in Europe which concluded that self-reported good health was the major facilitator of long working life among both women and men [2]. Other studies found that poor health and several psychosocial work characteristics, such as less possibility to develop skills and knowledge and low appreciation at work were associated with worse labour market attachment [42] as well as with higher risk of long SA and DP [10, 11, 13, 14].

Low education has also been shown to be associated with unstable working life trajectories [36]. However, in our study, higher education and adverse work environment were associated with the membership of a sequence cluster where active states were interrupted with longer parental-leave periods among women, which somewhat contradicts the findings of a previous Finnish study where lower education, lower attachment to the labour market, and immigrant background were the strongest predictors of long parental leave among women [37]. Another Swedish study also found that the use of parental-leave benefits was higher among immigrant women compared with native women [43] but this difference decreased along with time spent in Sweden, suggesting that lower attachment to the labour market was the main reason of longer parental-leave use among immigrant women [43]. The reasons for our different results might be related to a different baseline population (in our study, those in paid work) and a different outcome definition, as our study focused on complex, long-term working life patterns including unemployment and SA/DP instead of a single outcome, such as the number of parental-leave days. Being born in Sweden and living in Stockholm was a predictor of membership in the "Parental-leave periods" cluster among men, which suggests that while the Swedish parental-leave benefit policy aims to motivate more equal sex distribution in taking parental leave, there might still be educational, cultural, financial, or other policy-related factors which act against this [5, 39]. In sum, our study fits in the integrated life-course perspective proposed by Amick et al. [21], which provides a more dynamic approach of describing labour market and health experiences in working-age individuals [21].

## Strengths and limitations

The key strengths of our study are that we linked population-based survey data with annual and comprehensive nationwide register microdata [27, 44] and the long follow-up with minimal attrition, thus minimizing selection biases that can impact the findings of many other types of observational studies. Compared with other methods, sequence analysis allowed us to deal with the complexity of the future course of the working life.

One limitation of our study is that self-reporting in ULF/SILC survey might introduce some degree of bias due to misclassification. When available, we used both survey-based and register-based information but some information, such as health behaviors and work-related variables were only self-reported. Even though we conducted sensitivity analyses with different analytical parameters and methods which resulted in similar cluster structures, certain individuals might have ended up in different clusters when using different analytical decisions. However, the "Active", "Parental-leave periods", and "Retirement" clusters were consistently built up from the same or similar number of individuals and it was mostly the "Unemployment & SA/DP periods" and the "SA/DP periods" clusters that varied through the different clustering algorithms. Furthermore, the definition of states is also a simplified representation of working life sequences and could not fully capture the complexity of people's working life.

Our findings are generalizable to Sweden and, to some extent, to similar welfare and economic settings during the investigated time-period. Nordic countries have provided flexible and generous parental-leave benefits and child care for a long time now, and sex differences in employment are less pronounced than in other countries that provide more limited access to those. Furthermore, to be able to assess the contribution of working conditions, we limited our study sample to those who reported being in paid work at baseline and therefore, our results are not generalizable to individuals marginalized from the labour market or to those who had not entered the labour market.

## Conclusions

In a cohort of people initially in paid work in Sweden, we found that during the 15-year follow-up, the absolute majority (65%) of all individuals were uninterruptedly economically active (working or studying). About a third of them belonged to working life sequence clusters with active states combined with some interruptions. Women had a much greater likelihood than men of having a working life course characterized by employment interrupted with parental-leave and SA/DP periods but no sex difference were observed in a working life course characterized by a mixture of SA/DP and unemployment periods. These findings do not indicate that men did not take parental leave at all, but rather that during the follow-up of 15 years, fewer of them spent more than 183 days a year on parental leave benefits. However, the findings still suggest that there is a gender gap in the working life course even in Sweden where the Nordic welfare state model provides extensive and equal support to the labour market participation for women and men and promotes equal sharing of childcare.

## Supporting information

**S1 Fig. Selection of the study participants.**
(PDF)

**S2 Fig. Sequence index plots of different states of future yearly working life sequences over a 15-year period.**
(TIF)

**S3 Fig. Sequence index plots of different states of future yearly working life sequences over a 15-year period by sex.**
(TIF)

**S4 Fig. Hierarchical cluster tree of sequences of future yearly working life sequences over a 15-year period.**
(TIF)

**S5 Fig. Measures of cluster partitions quality for different numbers of sequence clusters.**
ASW: Average Silhouette Width HG: Hubert's Gamma, PBC: Point Biserial Correlation HC: Hubert's C.
(TIF)

**S6 Fig. Mean time spent in each sequence states by sequence clusters.**
(TIF)

**S7 Fig. Sequence index plots of four clusters of working life sequences.**
(TIF)

**S8 Fig. Sequence index plots of six clusters of working life sequences.**
(TIF)

**S9 Fig. Sequence index plots of seven clusters of working life sequences.**
(TIF)

**S10 Fig. Measures of cluster partitions quality for different numbers of sequence clusters by women and men.**
(TIF)

**S11 Fig. Sequence index plots of four clusters of working life sequences for women.**
(TIF)

**S12 Fig. Mean time spent in each sequence states by sequence clusters for women.**
(TIF)

**S13 Fig. Sequence index plots of four clusters of working life sequences for men.**
(TIF)

**S14 Fig. Mean time spent in each sequence states by sequence clusters for men.**
(TIF)

**S1 Table. Transition probabilities between different activity.**
(DOCX)

**S2 Table. Measures of cluster partitions quality for different numbers of clusters.**
(DOCX)

**S3 Table. Measures of cluster partitions quality for different numbers of clusters for women and men.**
(DOCX)

**S4 Table. Multinomial regression analysis for the associations between sex and different working life sequence clusters for alternative 6 clusters solutions.**
(DOCX)

**S5 Table. Associations between baseline factors and cluster membership among women.**
(DOCX)

**S6 Table. Associations between predictors and cluster membership among men.**
(DOCX)

## Acknowledgments

We thank to Annika Evolahti (Department of Clinical Neuroscience, Karolinska Institutet) for her editing support.

## Author Contributions

**Conceptualization:** Katalin Gémes, Katriina Heikkilä, Kristina Alexanderson, Kristin Farrants, Ellenor Mittendorfer-Rutz, Marianna Virtanen.

**Data curation:** Kristina Alexanderson, Ellenor Mittendorfer-Rutz, Marianna Virtanen.

**Formal analysis:** Katalin Gémes.

**Funding acquisition:** Ellenor Mittendorfer-Rutz, Marianna Virtanen.

**Methodology:** Katalin Gémes, Katriina Heikkilä, Kristina Alexanderson, Marianna Virtanen.

**Writing – original draft:** Katalin Gémes.

**Writing – review & editing:** Katalin Gémes, Katriina Heikkilä, Kristina Alexanderson, Kristin Farrants, Ellenor Mittendorfer-Rutz, Marianna Virtanen.

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
