## [Decision Letter · Decision Letter 0]

30 Aug 2022

PONE-D-22-14059Working life sequences over the life course among 9269 women and men in Sweden; a prospective cohort studyPLOS ONE

Dear Dr. Gemes,

Thank you for submitting your manuscript to PLOS ONE. After careful consideration, we feel that it has merit but does not fully meet PLOS ONE’s publication criteria as it currently stands. Therefore, we invite you to submit a revised version of the manuscript that addresses the points raised during the review process.

We look forward to receiving your revised manuscript.

Kind regards,

Renuka Sane

Academic Editor

PLOS ONE

Journal Requirements:

“This work was financially supported by the Swedish Research Council for Health, Working Life and Welfare, FORTE (2018-00547) and utilised data from the REWHARD consortium, supported by the Swedish Research Council (VR grant number. 2017-00624).”

Reviewers' comments:

Reviewer's Responses to Questions

**Comments to the Author**

1. Is the manuscript technically sound, and do the data support the conclusions?

Reviewer #1: Partly

Reviewer #2: Yes

2. Has the statistical analysis been performed appropriately and rigorously? 

Reviewer #1: Yes

Reviewer #2: Yes

3. Have the authors made all data underlying the findings in their manuscript fully available?

Reviewer #1: Yes

Reviewer #2: No

4. Is the manuscript presented in an intelligible fashion and written in standard English?

Reviewer #1: Yes

Reviewer #2: Yes

5. Review Comments to the Author

Reviewer #1: This study uses sequence analysis on a Swedish cohort created from registry data to create employment sequences for men and women. Sequence analysis is a useful tool for characterising life courses and is often used to characterise employment life courses. However, there are several major issues with the paper as it currently stands.

First, it is not clear what the rationale and motivation, or even the research question is here. The introduction begins with a focus on extended working lives but the sequences included are truncated at age 50 so it is not possible to contribute to this policy debate with this analysis. Perhaps the aim of the analysis is to describe diversity in employment life courses, but this hasn't really been achieved with only five groups capturing employment sequences for two genders. A secondary aim is to examine whether a range of factors are associated with employment sequences but there are no hypothesised directions of association nor any particular attempt to explain why these factors are important to investigate. Later in the statistical analysis section the authors state the regression models are to examine whether these covariates explain gender differences, but this is not developed in the introduction and would need careful thought to conceptually distinguish between confounders and mediators.

I suspect one reason for the lack of rationale and clear objectives is a lack of engagement with the literature. For example, the authors state that 'the contribution of working, living and health conditions to working-life participation among women and men has rarely been examined from a life course perspective. This could hardly be further from the truth. See studies by Lacey, McMunn, Berkman's group, Wahrendorf, Stone, Engels, Stafford, McDonough and others. There is a relatively large life course literature on this topic.

Which leads to my third comment which is the lack of engagement with key life course concepts such as timing of transitions, duration, accumulation, destandardisation, diversity etc (for one eg. see McMunn et al ALCR 2015).

In relation to the analysis conducted, the main issue is the lack of diversity in five groups which cover both men and women. There is nothing much of interest to report with these five groups. The purpose of sequence analysis is to illuminate life course patterns for different groups. The current analysis simply show that women have weaker ties to the labour market than men (which we know, even in Sweden) and results could have just as easily been conducted with count variables for the number of years in each state.

The authors go to lengths to justify their selection of five groups, but I was unfamiliar with many of the acronyms included in ST2 and they are not explained in the text. Neither are we given a clear description of the clusters in terms of Ns and gender % in each group, nor the duration of states and timing of transitions within each cluster. If the authors are interested in gender differences, they could consider running the cluster analysis separately for men and women and seeing how different the group solutions are between the two.

Small points:

What is SA/DP, this is never described.

Why have you mentioned cause of death data when you are not looking at mortality?

Some tables are lacking titles and a clear description of what numbers in them represent.

Having a similar number of people in each cluster is not a pre-requisite, nor even particularly useful, for determining the number of clusters.

ST4 is referred to in the text but I could not find it in the supplementary material.

Reviewer #2: This paper aims to analyze the gender gap in interruptions in working life sequences for individuals aged 18-50 in Sweden. The paper is very clearly and concisely written, authors have done a thorough exercise in terms of setting up the background, conducted a sound empirical analysis and provided a helpful discussion of their results. The research question assessed in the paper is a relevant one as Sweden is one of the developed nations with relatively smaller sex differences in employment/wage rates and progressive parental-leave options, these are social norms that many other nations wish to emulate. Hence a lot can be learned from the results of this paper, to be specific in spite of a generous parental-leave option which encourages equal sharing of parental responsibilities it is women who face higher number of interruptions in their working life sequences as represented by higher likelihood of women (vs men) belonging to “Parental-leave periods” and “Sickness Absence/Disability Pension” categories.

Below are my comments for the authors.

The analysis in the paper relies on a cohort dataset where 9269 individuals are followed up over a course of 15 years. The empirical analysis comprises of two parts. The first part comprises of cluster analysis (which groups similar working life sequences into 5 mutually exclusive categories) and the second part uses the information from part one to analyze the gender difference in the working life sequences (5 mutually categories from part one).

Major comments:

1) The cluster analysis results in 2-10 cluster solutions. And here I think researcher’s degree of freedom creep in. Authors chose a 5 cluster solution and provided the following short description “The measure of cluster partition quality (S2 Table, S4 and S5 Fig), in combination with the decision to keep the number of individuals within clusters on a reasonable level were used to decide the number of clusters to be the above-listed five.”.

I would recommend strongly to do two things:

a) Provide some more details about the choice being made i.e. a 5 cluster solution (instead of say a 4 or 6 cluster solution). Essentially the authors need to substantiate the critical choice they made with few objective arguments. For example, which measures of partition quality were used in this critical choice? To be precise, mention the values being higher or lower in comparison to other competing choices.

b) Since the analysis in part two, i.e. multinomial logit regressions (to assess gender gap) uses the cluster solution (mentioned above) as the dependent variable so the authors should ideally show that their results are robust to alternate choices of cluster solution. This would basically mean that Table 2 needs to be replicated for alternate cluster solutions. Looking at Table S2 Table it could mean using 3 cluster or 4 cluster or 6 cluster solution (instead of the original choice i.e. 5). This can be supplementary table but please at least 2 more cluster solutions (ideally one with higher of cluster solutions and another with lower value).

2) The detailed results related to separate results for men and women (Table S4 and S5) use sociodemographic variables from the baseline survey (the individuals from this survey were then followed up for next 15 years to create a trajectory of their working states). However not all of these sociodemographic characteristics are time invariant. For example, living alone with children variable (many respondents might not have had children at the time of survey but subsequently this status might have changed). Similar arguments hold true hours-per-week (as nature of work might have changed during the 15 year follow up period), higher education, health-related factors etc. Can the authors comment on this? Also if indeed these variables are time variant then the analysis should only be restricted to those variables which are time-invariant (and mention this as a limitation as well).

3) Provide more details about how the parental-leave policy is structured to provide equal parental benefits. This would go a long way in terms of explaining the huge gender difference which is observed for the parental-leave category (OR 33.2). This could also provide a helpful discussion which steers towards better policy design.

Minor comments:

1) Why does figure S2 have the highest value on Y axis as 8933, shouldn’t it be 9269?

2) All figures are pixelated, that is their appearance is not the best. It would be highly recommended to provide higher resolution images.

3) The manuscript refers to Table S4 in text but this table was missing from the manuscript (I could not locate a hyperlink for this)

4) Can authors provide very short (1-2 line) description about average at which males and females have their first child in Sweden, mean number of children, within country migration statistics (for example migration to and from Stockholm area. Would be helpful to know these details as some variables being used in the analysis are related to these).

5) The citation for reference number 2 has curly brackets (in the introduction) while all other citations have been put inside square brackets.

6. PLOS authors have the option to publish the peer review history of their article (what does this mean?). If published, this will include your full peer review and any attached files.

Reviewer #1: No

Reviewer #2: No

---

## [Author Response · Author response to Decision Letter 0]

13 Dec 2022

Point-by-point responses to the reviewers’ comments

Reviewer #1:

1. (1)”This study uses sequence analysis on a Swedish cohort created from registry data to create employment sequences for men and women. Sequence analysis is a useful tool for characterising life courses and is often used to characterise employment life courses. However, there are several major issues with the paper as it currently stands. First, it is not clear what the rationale and motivation, or even the research question is here.”

Authors’ response:

We thank the reviewer for the thorough review. We agree that the introduction was not clear enough and have now revised it to better reflect the study’s aims. Moreover, in line with your suggestions, we clarified the aim and rationale of the study and the hypotheses as following: “The specific research questions were to explore: 1) which types of overarching future working life sequence groups can be identified during a 15-year period in a cohort of individuals in paid work at baseline? 2) whether women have more often than men working life courses characterized by several interruptions, such as parental leave, SA/DP, or unemployment? 3) whether adjusting for baseline sociodemographic, health- and work-related factors weaken the possible associations between sex and the specific sequence cluster membership? 4) whether these factors are independently associated with the specific sequence cluster membership in a separate analysis among women and men?” (page 6)

2. “The introduction begins with a focus on extended working lives but the sequences included are truncated at age 50 so it is not possible to contribute to this policy debate with this analysis.”

Authors’ response:

We might have been unclear, however, to clarify: the sequences were not truncated at the age of 50. As we wanted to follow all people for 15 years, we included those who were 50 years or younger at inclusion, as the traditional age for old-age pension is 65 in Sweden. We now rephrased the first paragraph in the Introduction section, so the focus now is less on the old-pension age itself, as this paper was not intended to contribute to the policy debate on pension age, but more to map working life sequences in a population of those in paid work at baseline, and to investigate whether there are any important sex differences in working life sequences; and in that case, what they might be. 

3. “Perhaps the aim of the analysis is to describe diversity in employment life courses, but this hasn't really been achieved with only five groups capturing employment sequences for two genders.”

Authors’ response:

Our research aim was to map the overarching future working-life sequences in a cohort of individuals in paid work when included. This is now made clear in the aim of the revised manuscript. 

That we ‘only’ included individuals in paid work by default limited the observable complexity of working life sequences. Therefore, while we expected diversity in working-life sequences we did not anticipate a high level of complexity given our study population and the annual resolution of sequences. As our aim was to use the integrated life-course approach we defined labour market states on an annual basis to be able to integrate both family-, work-, and health-related interruptions of working life. Register data on sickness absence and disability pension have rarely been used in other types of studies. 

Actually, we regard the relative lack of complexity in the working-life sequences as one of the important findings of our study - as it suggests that the majority of both women and men have a working life course that is not interrupted by long breaks due to longer sickness absence, unemployment, or parental-leave periods. The mean time spent in an economically active state was close to 15 years, while the mean time spent in other states was less the 2 years– something that also reflects the limited diversity of working life sequences in this population. With this limited diversity in this population, a relatively low number of clusters could be expected. The chosen 5-cluster solution was a compromise between the best cluster partitioning, the highest possible complexity, and having sufficient power to be able to run the multinomial regression analysis. In the revised manuscript, following your comment, we now also present other cluster solutions in the supplement material (Fig S7, S8 and S9, Table S3, Table S3 and S4). Improvement in cluster partitioning measures over 5 clusters was minimal and using over 7 clusters they did not improve further at all. 

4. “A secondary aim is to examine whether a range of factors are associated with employment sequences but there are no hypothesised directions of association nor any particular attempt to explain why these factors are important to investigate. 

Authors’ response:

You are right, we had not hypothesised about the directions of associations. This was an exploratory observational study aiming to gain knowledge on the associations between employment sequence clusters and socioeconomic, health- and work-related factors such aspects. The choice of these factors was based on two considerations. One was based on which factors a large number of studies have shown are of importance for people’s work situation, sickness absence, etcetera. The other was more pragmatic: we could only include factors that were available, through the surveys and registers. In this study, our focus was to use survey data on the detailed work environment, including physical and psychosocial work environment, and information on subjective health, health behaviours, and chronic conditions. We have elaborated in the Introduction (on pages 4-5) about the selection of these factors.

5. Later in the statistical analysis section the authors state the regression models are to examine whether these covariates explain gender differences, but this is not developed in the introduction and would need careful thought to conceptually distinguish between confounders and mediators.”

Authors’ response:

Thank you for this comment, you are right and we have now modified this sentence in the Introduction as follows: “. To examine whether the associations between sex and cluster memberships attenuated, several sets of covariates were gradually included in the models.”. 

Our study design did not allow us to investigate any causal mechanism, or distinguish between confounders and mediators, but only to map the most common patterns examining health-, work-, and family-related interruption in working life course and their possible association with sex and to map the main factors that were associated with different working life sequence clusters in women and men. As we measured all such factors only at baseline, causal inference would not be possible with this study design. However, we agree with the reviewer that conceptualizing and quantifying the effect of confounders and mediators on working life courses in women and men is an important research question, and we would like to pursue it in future research.

6. (2)”I suspect one reason for the lack of rationale and clear objectives is a lack of engagement with the literature. For example, the authors state that 'the contribution of working, living and health conditions to working-life participation among women and men has rarely been examined from a life course perspective. This could hardly be further from the truth. See studies by Lacey, McMunn, Berkman's group, Wahrendorf, Stone, Engels, Stafford, McDonough and others. There is a relatively large life course literature on this topic.”

Author’s response:

Thank you for alerting us on not having referred to all the relevant literature on this topic. We are aware that these aspects have been researched by many important researchers in many different projects, and we should of course have mentioned these. What we aimed to state is that information on sickness absence and disability pension has seldom been included in studies using sequence analysis, the focus has rather been on work-family life courses and work participation of women in relation to different work- and health-related outcomes. We have now revised the Introduction and included the most relevant literature and clearer pointed out that sickness absence and disability pension were not included in the previous literature (page 5).

7. (3). ”Which leads to my third comment which is the lack of engagement with key life course concepts such as timing of transitions, duration, accumulation, destandardisation, diversity etc (for one eg. see McMunn et al ALCR 2015).”

Author’s response: 

Thank you for referring to this important paper. We clearly need to be more explicit when using the concepts. Therefore, we now refer to the concepts of “duration” and “diversity” in the Introduction as follows: “Some of the previous research focused on the examination of the inter- and intra-individual diversity of work-family life courses [16-21]. McMunn at al., [16] examined the changes in work-family life courses over time and showed that women’s work-family life courses converged to men’s while the diversity of the individual work-family life courses increased” referring to the Mc Munn et al ALCR 2015 paper. However, other concepts such as de-standardization and differentiation are not relevant to our study questions as we did not compare different time periods, but we mapped the sequences within the same cohort. Nor did we aim to focus on specific transitions or accumulation of exposure in this study. The aim of this study was descriptive, to map the most common working-life sequences and describe their diversity by forming groups of similar sequences. We agree with the reviewer; these concepts are important from the life-course perspective and can be studied specifically after this first step of explorative research.

8. (4.) “In relation to the analysis conducted, the main issue is the lack of diversity in five groups which cover both men and women. There is nothing much of interest to report with these five groups. The purpose of sequence analysis is to illuminate life course patterns for different groups. The current analysis simply show that women have weaker ties to the labour market than men (which we know, even in Sweden) and results could have just as easily been conducted with count variables for the number of years in each state.”

Author’s response: 

We agree that in this cohort of individuals in paid work at baseline, the diversity of life course patterns is limited. However, we regard this by itself as an important finding, showing that most both women and men in Sweden who are attached to the labour market actually have uninterrupted working life courses. This finding is important and has not been shown before in a large cohort with register-based information on working life states. As we have described above in Point 4, we chose sequence analysis to be able to capture different working life sequences with both health-, family-, and work-related interruptions using register microdata, which is one of the novel aspects in our study and required a method that can handle mutually exclusive labour market states. 

Therefore, we argue that methods based on count variables could not be able to capture this complexity as they could not handle several types of outcomes in one analysis. In addition, we prepared the study protocol, including the analyses, a priori, without knowing how much the sequences would differ from each other. We agree that several analytical approaches are important in studies of working life courses, and we see our study as one contribution in this area. 

9. “(5.) The authors go to lengths to justify their selection of five groups, but I was unfamiliar with many of the acronyms included in ST2 and they are not explained in the text. Neither are we given a clear description of the clusters in terms of Ns and gender % in each group, nor the duration of states and timing of transitions within each cluster. If the authors are interested in gender differences, they could consider running the cluster analysis separately for men and women and seeing how different the group solutions are between the two.”

Author’s response: 

We have now added the description of the acronyms to the tables. We also added to the Results section information on the different statistical measures we considered when deciding on the 5-cluster solutions (“The cluster partition quality measures showed the best fits for a five- and a seven-clusters solution (lowest point biseral correlation for five clusters and highest Hubert’s Gamma, Average Silhouette Width, and Calinszki-Habarasz indexes for the seven-clusters solution) but only with a minimal difference in indexes between the five- and seven-clusters solutions (Table S2, Fig S4, and Fig S5).”). Moreover, we now added the % of women in each sequence group beside the number of individuals, which was already presented in the Results section (page 12) . We also added a figure for each cluster for the mean time spent in each state (now Fig S6). This study did not aim to focus on the timing of specific transitions – that would have required other analytical methods in a study with other aims. Moreover, we added the suggested sex-stratified analyses as a supplement (now Fig S10, S11, S12, S13, S14, and Table S3) and added their description in the Method section: “To explore whether these cluster solutions were similar among women and men, we presented the results from cluster analysis stratified by sex.” and Results section “When the cluster analysis was stratified by sex, we found a four-clusters solution to have the best fit for women and a four- and five-clusters solutions to have the best fit for men (Table S3, Fig S10). We also present the four clusters for both women and men with mean time spent in each state in Supplement Material (Fig S11, Fig S12, Fig S13, and Fig S14).” 

10. Small points:

10.1 “What is SA/DP, this is never described.”

Authors’ response: 

Thank you for alerting us on this! We have now modified and expanded the Abstract and Methods section to clarify that SA/DP meant sickness absence/disability pension.

10.2 “Why have you mentioned cause of death data when you are not looking at mortality?”

Author’s response: 

Mortality was not an outcome of interest; however, we used the Cause of Death Register for information when defining the study cohort and identifying those who died during the follow-up and were censored. This is why we mention the use of this data source under the Methods section. However, we have now rewritten this paragraph to explicitly state which information we obtained from this register: “To identify those who died during the follow-up, we obtained the year of death from the Cause of Death Register held by the National Board of Health and Welfare.”

10.3 “Some tables are lacking titles and a clear description of what numbers in them represent.” 

Authors’ response: 

Thank you, we have now added titles and descriptions to all tables.

10.4 “Having a similar number of people in each cluster is not a pre-requisite, nor even particularly useful, for determining the number of clusters.”

Authors’ response: 

We agree with the Reviewer about this, and we did not use this as a criterion to determine the number of clusters. Instead, we used different measures of partition quality to decide on the best fit. The five- and seven-cluster solutions had very similar cluster partition measures, so we decided to present the five clusters and continue with the multinomial regression considering the feasibility of further analysis and the readability of the results. Now we added the following description of our decision steps under the Results: “The cluster partition quality measures showed the best fits for a five- and a seven-clusters solution (lowest point biseral correlation for five clusters and highest Hubert’s Gamma, Average Silhouette Width, and Calinszki-Habarasz indexes for the seven-clusters solution) but only with a minimal difference in indexes between the five- and seven-clusters solutions (Table S2, Fig S4, and Fig S5). To consider the feasibility and readability of the multinomial regression analyses we chose the five-sequence cluster solutions for further analysis (Fig 3). In the five-cluster solution, the mean time spent in each state is presented in Fig S6. Four-, six-, and seven-cluster solutions are presented in Fig S6, Fig S7, and Fig S8.”

10.5“ST4 is referred to in the text but I could not find it in the supplementary material.”

Authors’ response: 

We have now uploaded the missing table which now is referred to as Table S6.

---

## [Editor Report · Decision Letter 1]

17 Jan 2023

Working life sequences over the life course among 9269 women and men in Sweden; a prospective cohort study

PONE-D-22-14059R1

Dear Dr. Gémes,

We’re pleased to inform you that your manuscript has been judged scientifically suitable for publication and will be formally accepted for publication once it meets all outstanding technical requirements.

Kind regards,

Sreeram V. Ramagopalan

Academic Editor

PLOS ONE
---

## [Editor Report · Acceptance letter]

23 Jan 2023

PONE-D-22-14059R1 

Working life sequences over the life course among 9269 women and men in Sweden; a prospective cohort study 

Dear Dr. Gémes:

I'm pleased to inform you that your manuscript has been deemed suitable for publication in PLOS ONE. Congratulations! Your manuscript is now with our production department. 

Kind regards, 

on behalf of

Dr. Sreeram V. Ramagopalan 

Academic Editor

PLOS ONE